# An EMG Patch for the Real-Time Monitoring of Muscle-Fatigue Conditions During Exercise

**DOI:** 10.3390/s19143108

**Published:** 2019-07-14

**Authors:** Shing-Hong Liu, Chuan-Bi Lin, Ying Chen, Wenxi Chen, Tai-Shen Huang, Chi-Yueh Hsu

**Affiliations:** 1Department of Computer Science and Information Engineering, Chaoyang University of Technology, Taichung City 41349, Taiwan; 2Department of Information and Communication Engineering, Chaoyang University of Technology, Taichung City 41349, Taiwan; 3Biomedical Information Engineering Laboratory, University of Aizu, Aizu-wakamatsu City, Fukushima 965-8580, Japan; 4Department of Industrial Design, Chaoyang University of Technology, Taichung City 41349, Taiwan; 5Department of Leisure Services Management, Chaoyang University of Technology, Taichung City 41349, Taiwan

**Keywords:** electromyogram, patch, muscle fatigue, application program

## Abstract

In recent years, wearable monitoring devices have been very popular in the health care field and are being used to avoid sport injuries during exercise. They are usually worn on the wrist, the same as sport watches, or on the chest, like an electrocardiogram patch. Common functions of these wearable devices are that they use real time to display the state of health of the body, and they are all small sized. The electromyogram (EMG) signal is usually used to show muscle activity. Thus, the EMG signal could be used to determine the muscle-fatigue conditions. In this study, the goal is to develop an EMG patch which could be worn on the lower leg, the gastrocnemius muscle, to detect real-time muscle fatigue while exercising. A micro controller unit (MCU) in the EMG patch is part of an ARM Cortex-M4 processor, which is used to measure the median frequency (MF) of an EMG signal in real time. When the muscle starts showing tiredness, the median frequency will shift to a low frequency. In order to delete the noise of the isotonic EMG signal, the EMG patch has to run the empirical mode decomposition algorithm. A two-electrode circuit was designed to measure the EMG signal. The maximum power consumption of the EMG patch was about 39.5 mAh. In order to verify that the real-time MF values measured by the EMG patch were close to the off-line MF values measured by the computer system, we used the root-mean-square value to estimate the difference in the real-time MF values and the off-line MF values. There were 20 participants that rode an exercise bicycle at different speeds. Their EMG signals were recorded with an EMG patch and a physiological measurement system at the same time. Every participant rode the exercise bicycle twice. The averaged root-mean-square values were 2.86 ± 0.86 Hz and 2.56 ± 0.47 Hz for the first and second time, respectively. Moreover, we also developed an application program implemented on a smart phone to display the participants’ muscle-fatigue conditions and information while exercising. Therefore, the EMG patch designed in this study could monitor the muscle-fatigue conditions to avoid sport injuries while exercising.

## 1. Introduction

In recent years, people like to use a wearable device to monitor their bodies’ condition during exercise. The wearable devices include an exercise watch, or a band to measure heart rate and the number of steps they take [1,2,3]. People also like to use indoor sport equipment, such as a treadmill, elliptical trainer, or an exercise bike, to build and maintain muscle mass and strength for their health [4]. Moreover, isokinetic exercise using specialized equipment has been widely used for functional rehabilitation and assessment [5]. When people do isokinetic exercise, their leg muscles go through continuous dynamic contractions [6]. However, if people exercise too much, their muscles might get overtired or get injured. Therefore, the correct way to exercise properly is an important issue for both healthy and sickly people.

Muscle fatigue is defined as a loss of the required or expected force from a muscle, and has been a popular research topic for a long time [7]. It is well known that the power spectrum of the surface electromyography (sEMG) shifts to a lower frequency during a sustained muscle contraction. The spectral parameters, such as the mean frequency (MNF) and the median frequency (MF), are the manifestation of localized muscle fatigue [8,9]. According to most reports that studied muscle fatigue, the changes in muscle fiber propagation velocity and the firing rate of muscle fibers could affect the sEMG power spectrum [8]. When the muscle is doing isometric or isotonic contraction, the sEMG could be considered as a wide-sense stationary signal, and thus, its power spectrum could be obtained by common spectral estimation techniques. However, the sEMG signal would easily connect different types of noise together. Therefore, the sEMG signal has to delete the background noise before being able to detect muscle fatigue.

In some studies, the sEMG was considered as a nonstationary stochastic process, and the shift of the innervation zone always happens during isotonic muscle contractions [8]. Molinari [9] assessed the changed spectrum of the sEMG signal when the fatigue was happening during dynamic muscle contraction. Dingwell [10] calculated the relation between the localized muscle fatigue and the changes in movement kinematics. Tscharner [11] utilized wavelet analyses to evaluate the sEMG parameters while doing light exercise. Some nonlinear methods for the sEMG signal were used to assess muscle fatigue, such as the Lempel–Ziv complexity measure [12] and fuzzy approximate entropy [13]. González-Izal et al. reviewed some linear and non-linear methods to estimate muscle fatigue [14]. In those studies, they used a wired multi-channel measurement system and complex methods to analyze the sEMG signals. However, those measurement systems and analysis algorithms could not easily be applied to a wearable device for real-time monitoring of muscle-fatigue conditions.

Isezaki et al. [15] developed a sock-type wearable device to measure the distal EMG signal of the lower leg muscles. They only detected the active amplitude of the EMG signal. Xi et al. [16] used wearable sEMG sensors to monitor seven different activities during the day. Their purpose was to protect people from falling down when they get sick or dizzy. Chang et al. [4] also designed a Bluetooth wearable EMG sensor which only had one channel and its sampling rate was 2000 Hz. Liu et al. [17] used the measured sEMG signal from an EMG sensor to detect the degree of muscle fatigue with different decomposition methods, including wavelet transform, empirical mode decomposition (EMD), and ensemble empirical mode decomposition (EEMD). They found that intrinsic mode functions during muscle fatigue would change to a high frequency of the sEMG signal. In these studies, the wearable EMG sensors were all used to record the sEMG signals. The analyses of these signals were done off line.

Therefore, the goal of this study is to design an EMG patch to monitor the muscle-fatigue conditions in real time while exercising. An analog circuit with two electrodes was designed to acquire an EMG signal. The micro controller unit (MCU) of this EMG patch was an ARM Cortex-M4, which could measure the MF of the first intrinsic component of the EMD for the sEMG signal in real time. Moreover, a Bluetooth low energy (BLE) module was built into the EMG patch to transmit the real-time MF to a smart phone or a notebook. In order to verify the performance of the EMG patch, the real-time MF values were compared with the off-line MF values, measured by the computer system. The EMD algorithm was delivered from the National Central University through the computer system [18]. There were twenty participants in this experiment and they rode an exercise bicycle at different speeds. They did the experiment twice. Finally, we designed an application program (APP) that showed a Google map, the weather information, and the data of the EMG patch. When there is a team riding the bicycles, a cyclist can use this APP to monitor his own information and the information of his team mates in real time.

This paper is organized as follows. Section 2 describes the designed EMG patch, the EMD method, and the MF definition. The way to detect the number of muscle contractions from the sEMG signal is also described in this section. Section 3 presents the APP, and Section 4 presents the results of the data taken from the twenty participants. These results are discussed in Section 5, and the conclusion is in Section 6.

## 2. Methods

Figure 1 shows the block diagram of an EMG patch which includes an analog circuit, an MCU, a BLE module, a power circuit, and an alarm system. We used a wireless technique to charge the battery. The alarm system has two LEDs to display the power and the status of the BLE module, and a device that vibrates to notify the user of over tiredness. In this study, the EMG patch had an accelerometer. However, it did not work in this study. In order to verify the real-time performance of an EMG patch, a computer-based measurement system was designed to analyze the MF values of the sEMG signal, off line. Figure 2 shows the framework of the comparison between the real-time EMG patch and the off-line computer system. The sEMG signal was recorded simultaneously through the EMG patch and a physiological measurement system. The real-time MF values were transferred to the computer through the BLE module, and then recorded. These values were compared with the synchronous off-line MF values measured by the computer system. Moreover, the amplitude of the sEMG signal could be detected and used to calculate the number of muscle contractions. The number of muscle contractions represents how fast they pedaled. The real-time degree of muscle fatigue and pedaling rate would be shown in an application program (APP).

### 2.1. Hardware of the EMG Patch

Figure 3 shows the analog circuit for the EMG patch. The raw sEMG signal is a low-amplitude signal, therefore it needed to be amplified. An instrument amplifier (AD8236, Analog Devices (AD) Company, Norwood, MA, USA), with a gain of 10, was used to enhance the signal. Because a virtual-ground technique was used, only two electrodes were needed to be used on the input terminals of the instrument amplifier. Operational amplifiers (AD8609, AD Company, Norwood, MA, USA) were used to design the filters, the amplifier, the peak rectifier, and the baseline offset circuit. According to a previous study [17], a two-order Butterworth high-pass filter (cutoff frequency 33.9 Hz) was used to remove the direct current (DC) offset and the baseline wandering, and a two-order Butterworth low-pass filter (cutoff frequency 482.5 Hz), was used to reduce the high-frequency noise and to avoid aliasing of the sampling signal. The gain of the non-inverting amplifier was 100. Finally, the baseline of the sEMG signal was raised to 1 V by a baseline offset circuit.

In the power circuit, the Texas Instruments (TI) wireless receiver chip (BQ51003YFPR TI company, Dallas, TX, USA) was used to charge the lithium battery. A voltage regular (TPS78233, TI company, Dallas, TX, USA) was used to provide a 3.3 voltage (V), and its input voltage was from the lithium battery. A negative voltage regular (TPS60400, TI company, Dallas, TX, USA) was used to provide a −3.3 V, and its input voltage was 3.3 V. The maximum power consumption of the EMG patch was 39.5 mAh.

The BLE module was the NrfF51822 (Nordic Semiconductor, Trondheim, Oslo, Norway). The STM32L432KC (STMicroelectronics, Geneva, Switzerland) is a 32 bit MCU which has a 64 kB static random access memory (SRAM), and a 256 kB flash memory. It uses a clock of 32 MHz. The power consumption is 84 uA/MHz when running at full capacity. The sampling rate of the EMG patch was 1000 Hz. Moreover, an electronic vibrator was used in the EMG patch to alert the user when the degree of muscle fatigue exceeds the designed threshold. Figure 4 shows two views of the EMG patch; (a) from the top, (b) from the bottom. The size of the main board is 30 mm, and the size of the electrode’s connector is 25 mm.

### 2.2. Empirical Mode Decomposition

EMD is applied to decompose an sEMG signal to detect muscle conditions. Details of the algorithm can be found in reference [17]. The pseudo code of the EMD algorithm is shown in Algorithm 1.

**Algorithm 1:** Pseudo code of the EMD algorithm**Input:** Given a signal *x*(*t*)  1. Set *r*(*t*) = *x*(*t*) and *k* = 0.  **while**
*r*(*t*) is not monotonous **do**
    2. Set *m*(*t*) = *r*(*t*).    **while**
*m*(*t*) is nontrivial **do**
      3. Interpolate the local minima and maxima, ending up with lower and upper envelopes, *e_min_*(*t*) and *e_max_*(*t*).      4. Compute the average $m\left(t\right)=0.5\left(e_{min}+e_{max}\right).$
      5. Extract the different signal d(t)=r(t)−m(t), and denote *d*(*t*) as *r*(*t*)    **end while**    6. Set *k* = *k* + 1.    7. Set $IMF_{k}\left(t\right)=d\left(t\right).$
    8. Set *r*(*t*) = *x*(*t*) − $\sum_{i=1}^{k}IMF_{i}\;\left(t\right)\;$
  **end while**
**Output.***x*(i)= $\sum_{i=1}^{k}IMF_{i}\;\left(t\right)\;+r\left(t\right)\;\;\;$

In this study, the size of the sampling data segment was 4096 points for each processing session, and the overlap points were 2048 points. According to the study of Liu [17], we only used the IMF1 component to measure the MF values. In order to run the EMD algorithm in an MCU system, we did Step 1 to Step 5 ten times. A second order polynomial function was used to get the upper and lower envelopes of signal on Step 3. However, on the computer system, the iteration of the IMF1 would end when the mean of the sum of the IMF1 was zero, and the third order function was used to estimate the upper and lower envelopes of the signal. Figure 5 shows the results of the EMD algorithm in the EMG patch; (a) the raw EMG, (b) the IMF1 of the sEMG, (c) the upper (red) and lower (blue) envelopes of the sEMG. The upper and lower envelopes obtained by our method are very close to the truth envelopes.

### 2.3. Median Frequency

Fast Fourier transform (FFT) was applied to the IMF1. After that, the MF was defined as the frequency where the accumulated spectrum energy is half of the total spectrum energy, as shown in Equation (1). Where the is the p(f) power spectrum density (PSD) of the IMF 1:(1)$$\int_{0}^{MF}p\left(f\right)df=\frac{1}{2}\int_{0}^{\infty }p\left(f\right)df$$

Then, we used the root-mean-square difference between the real-time and the off-line MF values, *E_RMS_*, to verify the performance of the EMG patch:(2)$$E_{RMS}=\sqrt{\frac{1}{N}\overset{N}{\underset{k=1}{\sum }}{\left(MF_{real-time}\left[k\right]-MF_{off-line}\left[k\right]\right)}^{2}}$$
where *MF_real-time_* is the real-time MF value, *MF_off-line_* is the off-line MF value, *N* is the samples of MF.

### 2.4. Detecting the Number of Muscle Contractions

When the muscle is performing dynamic contractions, the amplitudes of the EMG signal have significant changes. Thus, the envelope of the EMG amplitudes can be used to determine the number of muscle contractions which represents the pedaling rate, the revolution per minute (RPM), while riding the bicycle. First, the EMG signal is downsampling:(3)$$x_{1}\left[n\right]=x\left[4n\right].$$
Then, a two-order Butterworth low-pass filter with a cutoff frequency of 90 Hz was used to remove the sEMG signal:(4)$$\frac{X_{2}\left(z\right)}{X_{1}\left(z\right)}=\frac{0.8066+1.6012z^{-1}+0.8006z^{-1}}{1+1.561z^{-1}+0.6413z^{-2}}.$$
The absolute value of the different filtered signal was used to extract the envelope of the muscle activity:(5)$$x_{3}\left[n\right]=x_{2}\left[n\right]-x_{2}\left[n-1\right],$$
(6)$$\;\;\;x_{4}\left[n\right]=\left|x_{3}\left[n\right]\right|.$$
Then, the integration was done twice:(7)$$x_{5}\left[n\right]=\overset{149}{\underset{k=0}{\sum }}x_{4}\left[n-k\right],$$
(8)$$x_{6}\left[n\right]=\overset{249}{\underset{k=0}{\sum }}x_{5}\left[n-k\right].$$

A peak marker with a size of 21 points was used to detect the peak of the envelope. Figure 6 shows the envelope (blue) of the EMG (black) signal which has very clear peaks. Thus, the peak could be easily detected. After that, the amount of peak within two seconds represents the pedaling rate.

### 2.5. Experiment Protocol

There were twenty healthy participants in this study consisting of 10 males and 10 females, aged from 20 years to 26 years. Before the collection of the data, all the participants were notified of the experiment protocols, and signed consent forms. An exercise bicycle, a Giant Taiwan, was used in the experiment. The EMG patch was worn on the gastrocnemius, as shown in Figure 7. We avoided using the belly position of the gastrocnemius muscle and shifted the electrodes to a higher position on this muscle. The Ag/AgCl electrodes (Kendall, SK, Canada) were used for the EMG recording, with a 10 mm diameter and self-adhesive supports. The positions of the electrodes for each subject were recorded, and the electrodes were placed on the same position in the subsequent experiments. Before wearing the EMG patch, alcohol was used to clean the surface to decrease any contact impedance. There were two riding speeds in this experiment, 60 RPM and 100 RPM. The 60 RPM represented light exercise, and the 100 RPM represented heavy exercise. The exercise bicycle could display the RPM on a smart phone. A 10 min session was required for each speed. In the pre-experiment, the participants rode at a specific speed and tried to keep that speed for at least 10 min. In order to show the reliability of the EMG patch, participants were requested to do the experiment twice, and they got three to four days to rest between the two experiments. A multi-channel physiological measurement system (KL-710, K&H MFG. CO. LTD., Taipei, Taiwan) was used to record the sEMG signal synchronously, the bandwidth was 30 Hz to 500 Hz, and the sampling rate was also 1000 Hz.

## 3. Application Program for the Cyclists

In this study, we developed an APP using a JavaScript program for the cyclists to use. The user could watch their own muscle conditions and the information of the other cyclists, the weather information, the GPS position, the pedaling rate, and the speed velocity of the bicycle on this APP. The EMG patch with the BLE module transferred the level of muscle fatigue and the pedaling rate to the APP every two seconds. Figure 8 shows the home page of the APP. In the beginning, the APP would show the addresses of the median address control (MAC) of all BLE modules. Then, the APP would connect the EMG patch automatically. When users touched the ‘map’ icon, a Google map would be displayed. When users touched the ‘stop’ icon, the APP would disconnect from the EMG patch.

When the APP got the ten MF values, the average of the first five MF values would be the baseline, and the average of the other MF values was used to calculate the level of muscle fatigue. If the averaged MF value was larger than the baseline value, the averaged MF value was replaced as the baseline value. Therefore, the baseline MF value must be the maximum value when estimating muscle fatigue. We defined 10 levels for the muscle-fatigue conditions. The zero level represented no muscle fatigue, and the tenth level represented the worst muscle fatigue. According to the “Detecting the Number of Muscle Contractions” section, the pedaling rate was also shown on this page. The level of muscle fatigue, $Level_{Muscle\_Fatigue}$, is defined below:(9)$$Level_{Muscle\_Fatigue}={\left[\frac{Baseline-MF_{avaerage}}{Baseline}\times 100\right]}_{10}$$

The APP data would be transferred to the server by 4G communication, which includes the $Level_{Muscle\_Fatigue}$, using one byte, the step velocity using one byte, and the GPS position using eight bytes. The GET method of the HTTP was used to transfer this data. According to the data of the GPS position, the speed of the bicycle would be calculated on the server. The server also got the weather information from the central weather bureau. Then, the data for each cyclist would be transferred back to the APP. Table 1 shows the data size and protocol. The total bytes are 23 bytes. Number 1 represents the data of the APP user. Number 2 represents the data of the others.

## 4. Results

Figure 9 shows the results of the EMD by the MATLAB code on the computer system. The activities of the muscle during the isotonic contractions are clearly displayed in the IMF1 which is the same as the study of Liu [17]. All the other IMFs and the residual signals do not show the clear activities of the muscle. Figure 10 shows the change of the real-time (black line) and the off-line (red line) MF values during the exercise. After ten minutes, the MF values have a significant shift towards a lower frequency. The slopes of their linear regression are −0.0248 Hz/s and −0.0247 Hz/s for the real-time MF values (short dash line) and off-line MF values (long dash line). The *E_RMS_* is only 3.68 Hz. Table 2 shows all *E_RMS_* between the real-time MF values and the off-line MF values for the twenty participants. We found that the worst *E_RMS_* was 4.85 Hz, and the best *E_RMS_* was 1.90 Hz in these experiments. For the first experiment, the average *E_RMS_* were 3.00 ± 1.11 Hz and 2.73 ± 0.53 Hz, for the male and female subjects. For the second experiment, the average *E_RMS_* were 2.40 ± 0.18 Hz and 2.72 ± 0.62 Hz, for the male and female participants. Table 3, Table 4 show the slope values of the MF changes for all participants. The slope values are all negative.

Figure 11 shows the map information of the APP. The red symbol represents the user, the blue symbol is another cyclist. The weather information shows on the top. The APP could help the user to understand the condition of their muscles and the riding information immediately. Moreover, they can also see the riding information of another cyclist. Therefore, cyclists could use this APP to ride the bicycle at an optimal speed for exercise.

## 5. Discussion

The term ‘consumer health care’ is a new concept in the industry, and it means that there is increased health awareness from using the wireless technique, helping to build healthy consciousness for every citizen [19]. There are some wearable devices developed to monitor the body’s physiological information, such as an electroencephalogram, an electrocardiogram, an EMG, or that show the respiration and heart rate [20]. These devices monitor the raw signals with low sampling rates, and their algorithms are not complex enough to get certain important features of the measuring signals [21]. Their advantages are that they have a long-term monitoring of about two to seven days [20]. However, if the device is only used to monitor the body condition in real time during exercise, it will only last one day or even shorter. Therefore, when the wearable device has to have a higher sampling rate and a higher complex algorithm, the way to design the firmware of the device would be a challenge.

Some wearable devices used the textile electrodes to measure electrocardiogram and EMG signals [16,19]. In this study, we designed bicycle pants which had two textile electrodes on the lining of the pants to measure the lateral femoral muscle, as shown in Figure 12a. When the subject wore these bicycle pants, the EMG patch was placed on the pants as shown in Figure 12b. However, the tail end of pants would follow the pedaling cycle to move. The textile electrodes also were moved. Therefore, the sEMG signal was coupled with many artificial noises. The sensitivity of assessing muscle fatigue became low. Thus, we used the Ag/AgCl electrodes to measure the sEMG signal, which also could hold the EMG patch on the muscle.

The limitations of wearable devices include the power consumption (mAh), the memory size (kilobytes), and a clock (MHz/s). In this study, the sampling rate of the sEMG signal was 1000 Hz. One segment was 2048 sampling points. The SRAM of the MCU only has 48 kilobytes for registering the data, because 16 kilobytes of memory are used for the other registers of the MCU. In the EMG patch, the analog-to-digital converter (ADC) buffer for 1024 sampling points needed 2 kilobytes of memory. Because the FFT and EMD methods belong to the complex algorithms, they need a large amount of memory. The multiplication numbers were $1024\times log_{2}2048$, and the additional numbers were $2048\times log_{2}2048$ for the FFT calculation [22]. The total memory size of the FFT algorithm for the real and complex numbers was 16 kilobytes. In the EMD algorithm, the memory size of one sampling data segment for the sEMG data was 8 kilobytes. The memory sizes of the registers for the upper and lower envelopes were 8 kilobytes and 8 kilobytes, separately. According to the Nyquist frequency, the first 1024 points of the PSD for the IMF1 component were used to estimate the MF value. Thus, the memory size for the PSD was 2 kilobytes. Because the four second data were used to estimate one MF value, one was used for the new data and the other one for the old data. Therefore, the total 46 kilobytes of SRAM were used to run the FFT and EMD algorithm.

In the EMD algorithm, the difference signal, d(t) [23] was obtained by the iteration method. The different IMF components were calculated in sequence. Since the running time of the EMD algorithm depends on the complexity of the signal, the EMD is very hard to be implemented in a real-time system. However, according to the study of Liu [17], the IMF1 of the sEMG signal had a higher sensitivity for detecting muscle fatigue than the other IMF components. We also found that the average value of the difference signal would be close to zero when its iteration time was ten. In Table 2, we compared the real-time MF values from the EMG patch and the off-line MF values with the computer system. Their *E_RMS_* values are very small, the average of which is only 2.17 ± 0.7019 Hz. This result shows that the IMF1 component can be obtained by iterating ten times, and the upper and lower envelopes were obtained by a second-order function. The running time for calculating an MF value was 0.213 s under the MUC with the 32 MHz clock.

In this study, the EMG patch was limited by the size of SRAM and the running clock of the MCU. Thus, we designed the sampling rate of the EMG patch to be 1000 Hz and the bandwidth of the anti-aliasing filter to be 33.9 to 482.5 Hz. Although the high cutoff frequency was very close to the Nyquist frequency of 500 Hz, the signal energy at the cutoff frequency has been attenuated to the half energy in the pass-band. Thus, the sampling EMG signal might have the aliasing phenomenon. However, according to the study of Liu [17], the maximum MF value for the EMG signal was about 280 Hz when the sampling rate was 2000 Hz and the sample data was 30 s. In our study, the maximum MF value was about 240 Hz when the sampling rate was 1000 Hz and the sample data was 4 s. The two results were very close. In Table 3 and Table 4, the slope values of MF change are all negative, which is similar to the results of the previous study [17].

## 6. Conclusions

Finally, we designed the EMG patch to monitor the conditions of muscle fatigue in the muscle during isotonic contraction. It used two electrodes to measure the sEMG signal. The ARM Cortex-M4 processor could run the FFT and EMD algorithm to detect the MF values of the EMG signal in real time. The results of this study show that the real-time MF values measured by the EMG patch were very close to the off-line MF values measured by the computer system. Moreover, an APP was designed in this study, which could display the levels of muscle fatigue and the riding information of the user, and the information of other cyclists in real time. Therefore, the EMG patch could be used to monitor the muscle-fatigue conditions during exercise to avoid sport injuries in the future.

## Figures and Tables

**Figure 1 sensors-19-03108-f001:**
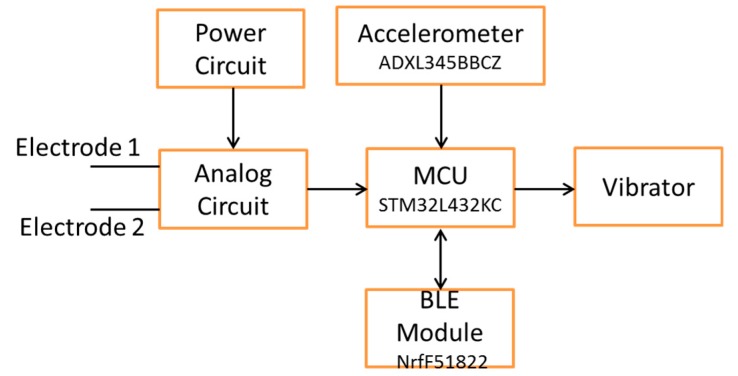
Block diagram of electromyogram (EMG) patch. MCU: micro controller unit; BLE: Bluetooth low energy.

**Figure 2 sensors-19-03108-f002:**
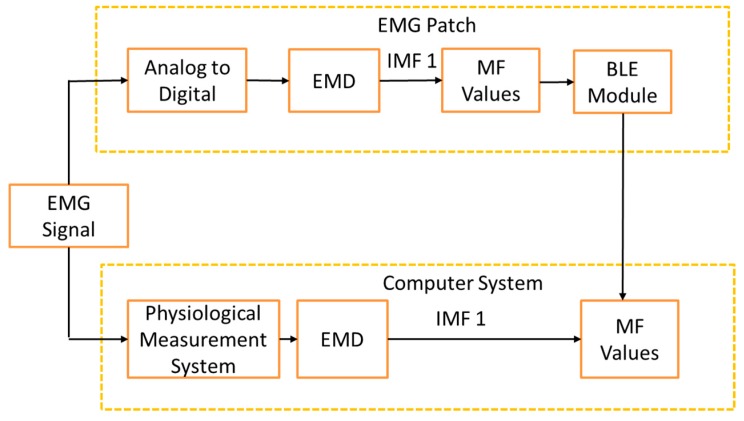
The framework of comparison between the real-time EMG patch and the off-line computer system. EMD: empirical mode decomposition; MF: median frequency.

**Figure 3 sensors-19-03108-f003:**
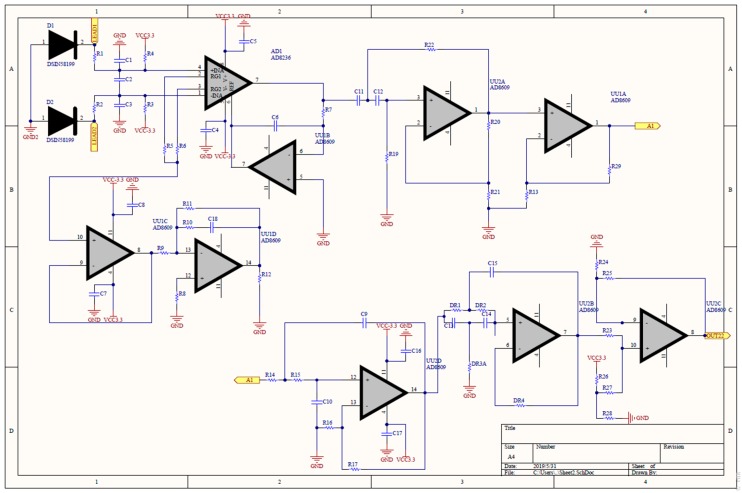
Analog circuit of the EMG patch.

**Figure 4 sensors-19-03108-f004:**
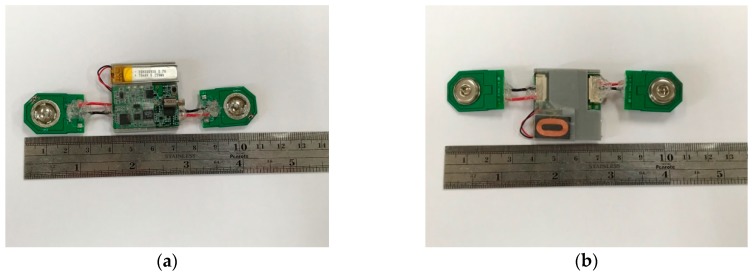
Two views of the EMG patch, (**a**) top view; (**b**) bottom view.

**Figure 5 sensors-19-03108-f005:**
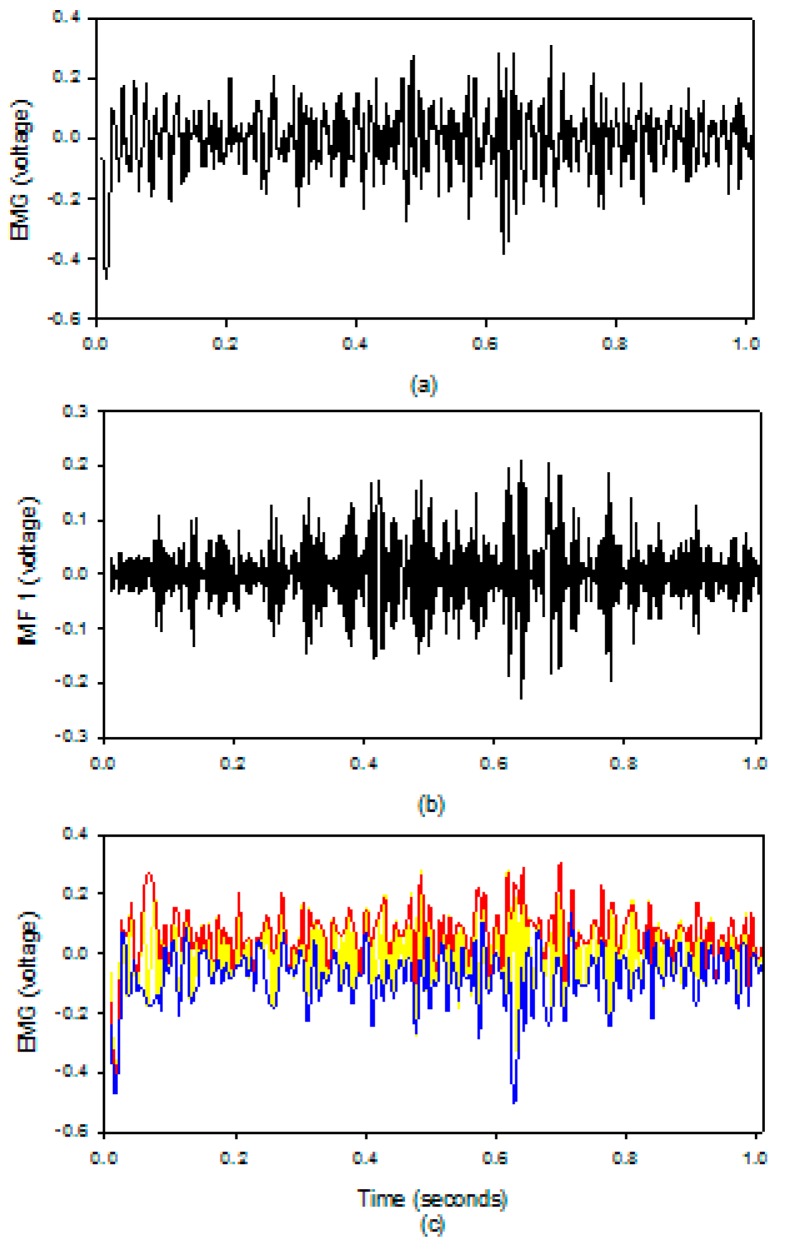
The results of the EMD algorithm on the EMG patch; (**a**) the raw EMG, (**b**) the IMF1 of the EMG, (**c**) the upper (red) and lower (blue) envelopes of the EMG, and the middle (yellow).

**Figure 6 sensors-19-03108-f006:**
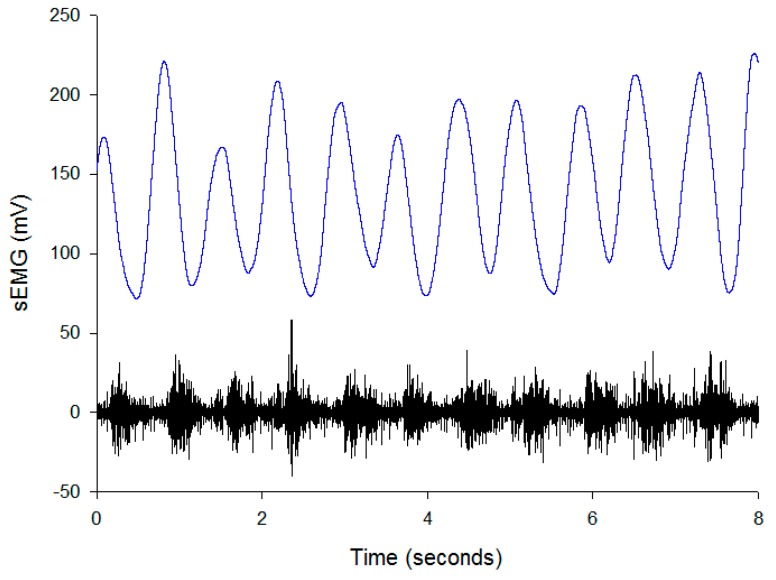
Surface electromyography (sEMG) (lower) and its envelope signals (upper).

**Figure 7 sensors-19-03108-f007:**
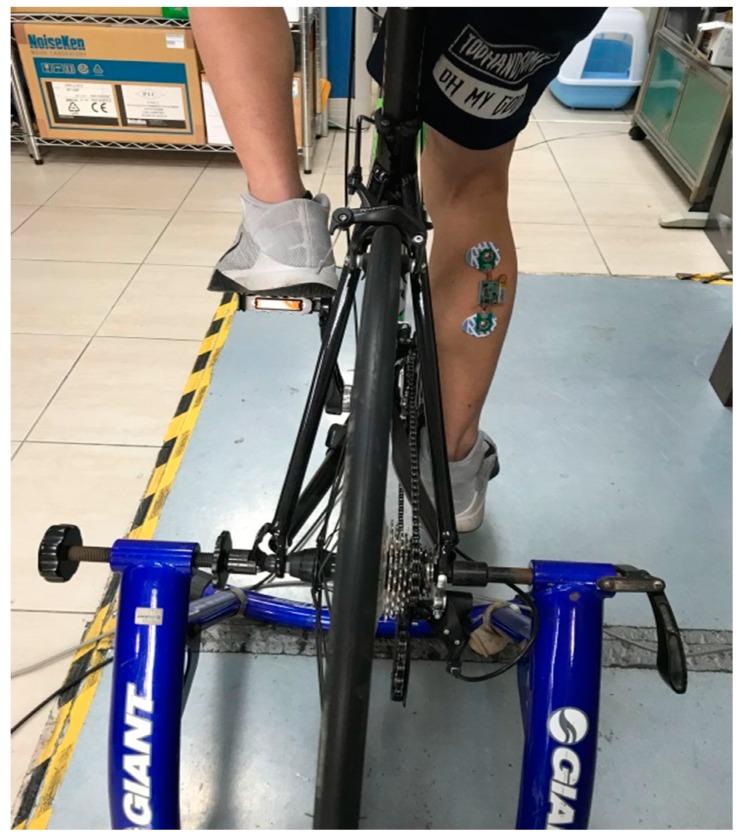
A photo of the experimental EMG patch.

**Figure 8 sensors-19-03108-f008:**
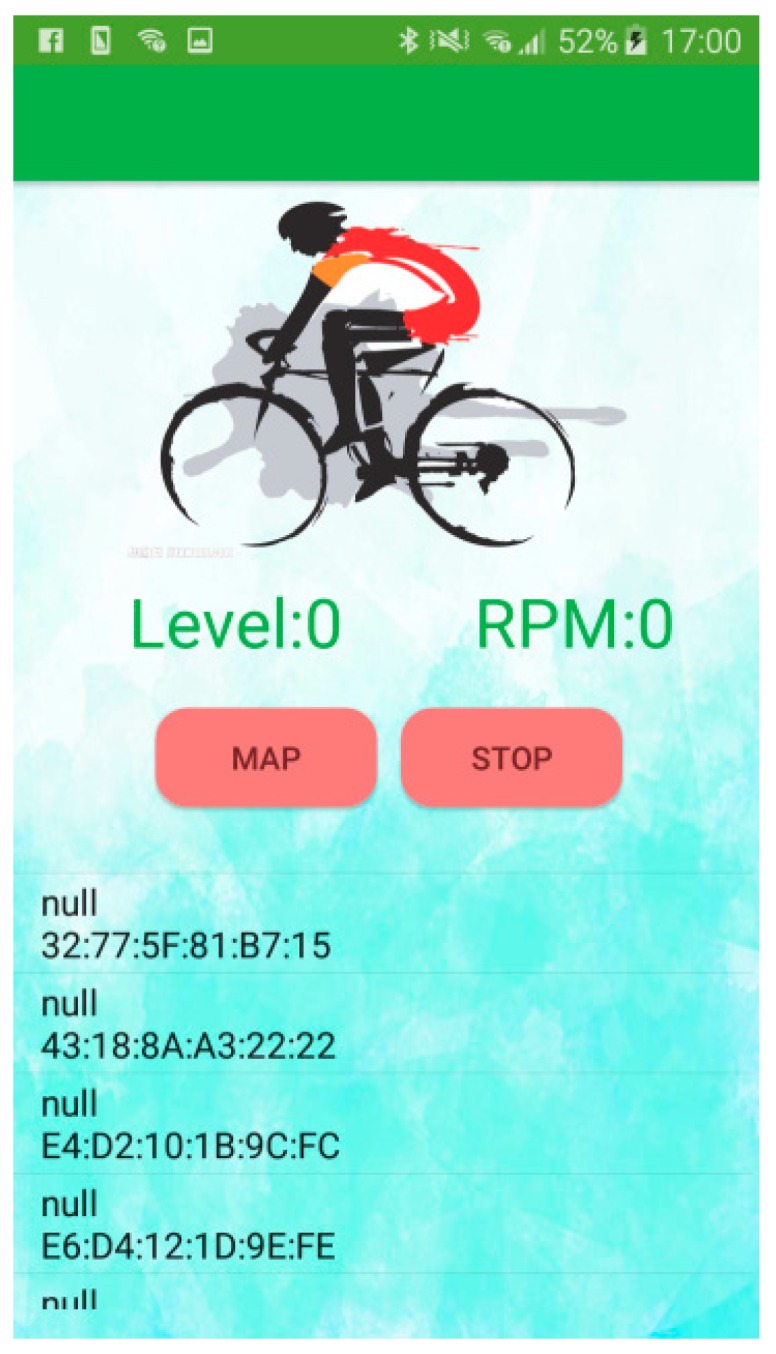
APP screen snapshot.

**Figure 9 sensors-19-03108-f009:**
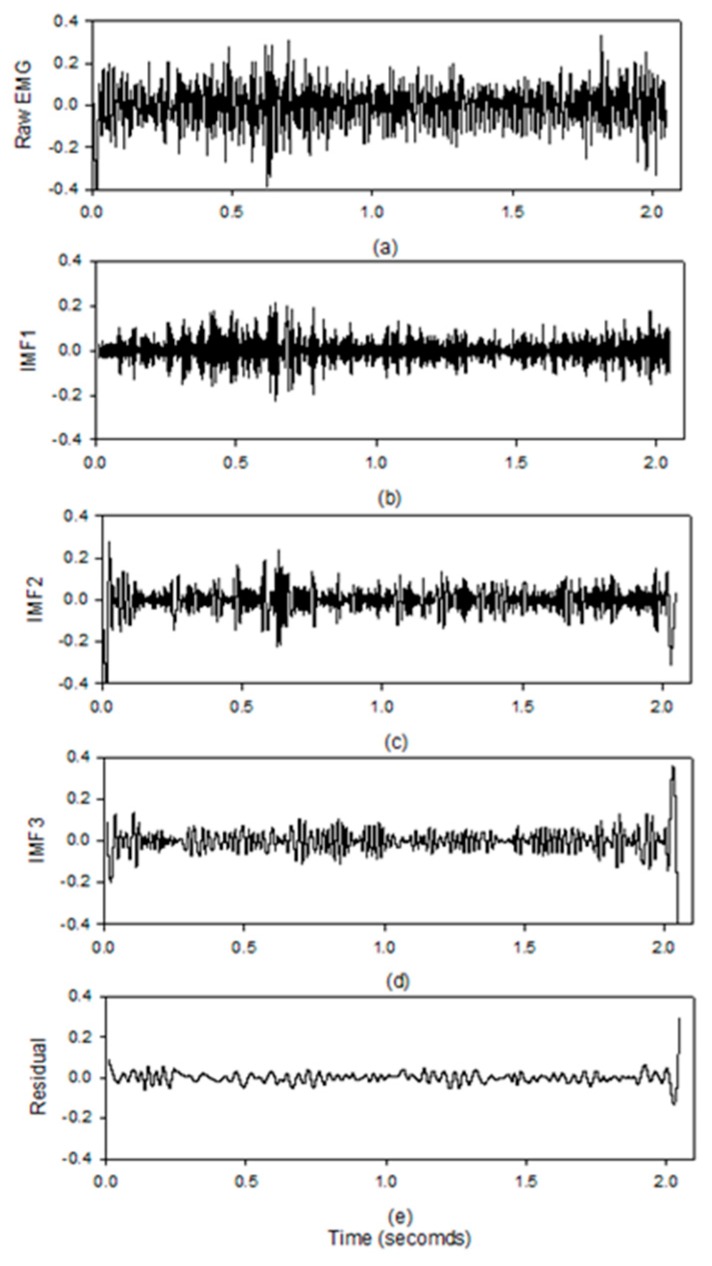
The IMFs of EMD for the sEMG signal decomposed using the MATLAB code in the computer system; (**a**) the EMG; (**b**) IMF1; (**c**) IMF2; (**d**) IMF3; (**e**) residual signal.

**Figure 10 sensors-19-03108-f010:**
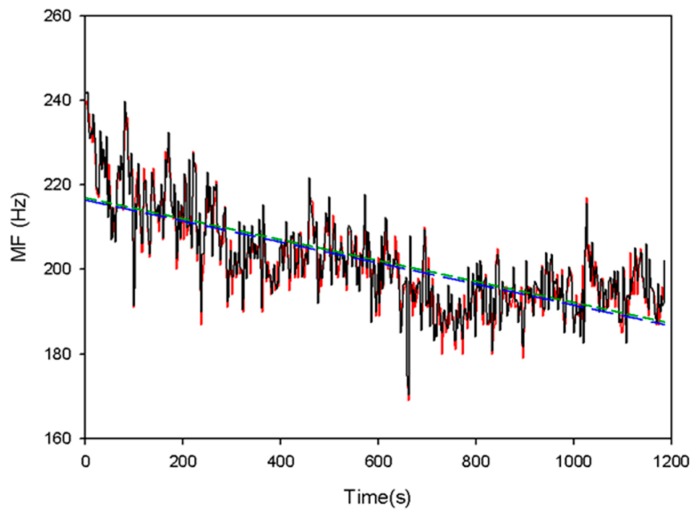
The real-time MF values on the EMG patch (red solid line) and the off-line MF values on the computer system (black dotted line). The slopes of their linear regression are −0.0248 Hz/s and −0.0247 Hz/s for the real-time MF values (short dash line) and off-line MF values (long dash line).

**Figure 11 sensors-19-03108-f011:**
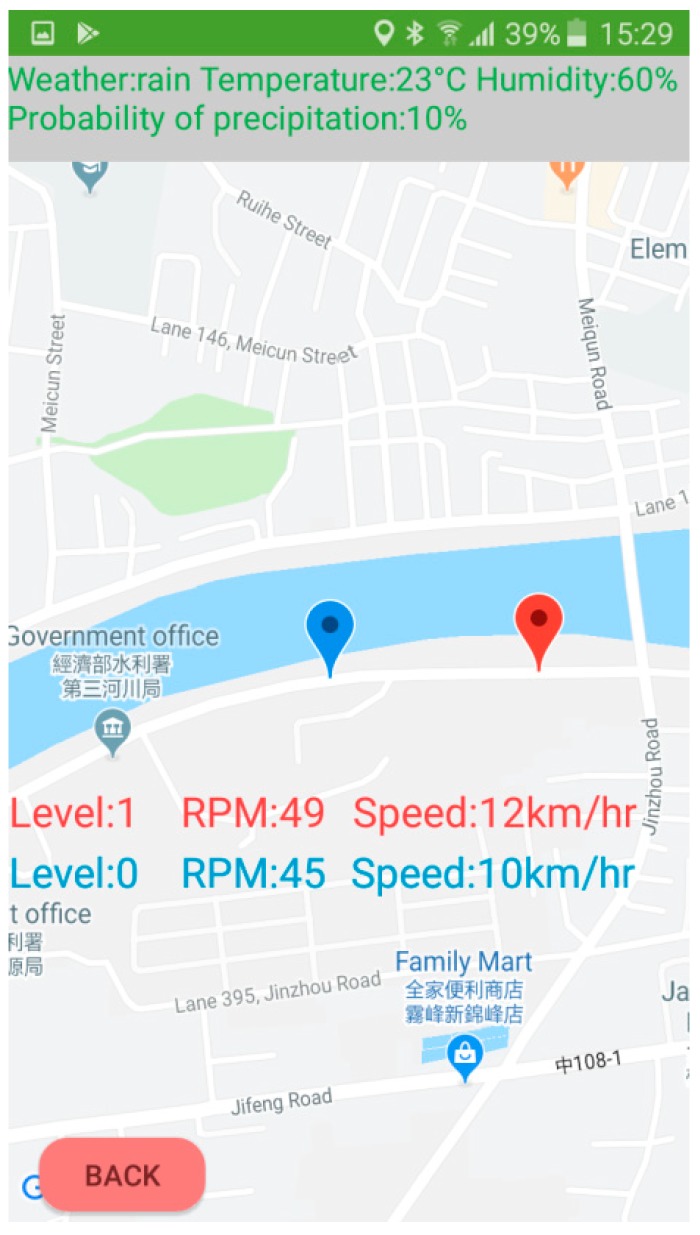
A sample map of the APP for a cyclist.

**Figure 12 sensors-19-03108-f012:**
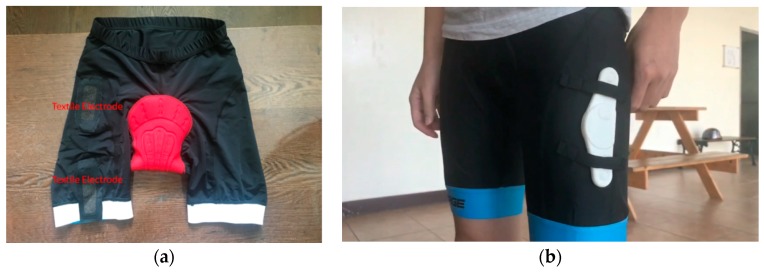
(**a**) Photo of the lining of the bicycle pants; (**b**) the EMG patch was placed on the bicycle pants to measure the lateral femoral muscle.

**Table 1 sensors-19-03108-t001:** The data size and protocol when data from the server is transferred back to the APP.

1 byte	2 bytes	2 bytes	2 bytes	2 bytes	2 bytes	2 bytes	2 bytes	8 bytes
No.	Level	Step’s velocity	Riding velocity	Weather	Temp	Humidity	Raining rate	GPS

**Table 2 sensors-19-03108-t002:** All *E_RMS_* values between the real-time MF value and the off-line MF values for the twenty subjects.

Number	$\mathrm{Male}\;(n=10)\;E_{RMS}\;(\mathrm{Hz})$		$\mathrm{Female}\;(n=10)\;E_{RMS}\;(\mathrm{Hz})$	
	First	Second	First	Second
No. 1	2.22	2.18	2.74	2.67
No. 2	4.85	2.18	2.74	2.67
No. 3	2.21	2.33	2.43	2.50
No. 4	2.79	2.45	2.21	1.90
No. 5	2.29	2.60	2.74	2.80
No. 6	2.01	2.40	3.10	2.51
No. 7	2.57	2.16	2.16	2.43
No. 8	4.56	2.60	2.22	2.94
No. 9	4.30	2.40	2.93	2.42
No. 10	2.17	2.23	3.91	4.29
Sum	3.00 ± 1.11	2.40 ± 0.18	2.73 ± 0.53	2.72 ± 0.62

**Table 3 sensors-19-03108-t003:** All slope values of MF change for the ten male subjects.

Male (n = 10)						
Number	First			Second		
	Real-Time MF (Hz/s)	Off-Line MF (Hz/s)	Error (Hz/s)	Real-Time MF (Hz/s)	Off-Line MF (Hz/s)	Error (Hz/s)
No. 1	−0.0049	−0.0050	0.0001	−0.0062	−0.0065	0.0003
No. 2	−0.0057	−0.0053	−0.0004	−0.0111	−0.0111	0
No. 3	0.0003	0.0003	0	−0.0063	−0.0059	−0.0004
No. 4	−0.0067	−0.0065	−0.0002	−0.0143	−0.0142	−0.0001
No. 5	−0.0059	−0.0061	0.0002	−0.0115	−0.0113	−0.0002
No. 6	−0.0025	−0.0027	0.0002	−0.0041	−0.0042	0.0001
No. 7	−0.0053	−0.0052	−0.0001	−0.0064	−0.0065	0.0001
No. 8	−0.0191	−0.0191	0	−0.0171	−0.0172	0.0001
No. 9	−0.0023	−0.0025	0.0002	−0.0029	−0.0029	0
No. 10	−0.0129	−0.0129	0	−0.0248	−0.0247	0
Sum	−0.0065 ± 0.00562	−0.0065 ± 0.00560	−0.0000 ± 0.00018	−0.0105 ± 0.00680	−0.0105 ± 0.00679	−0.0000 ± 0.00020

**Table 4 sensors-19-03108-t004:** All slope values of MF change for the ten female subjects.

Female (n = 10)						
Number	First			Second		
	Real-Time MF (Hz/s)	Off-Line MF (Hz/s)	Error (Hz/s)	Real-Time MF (Hz/s)	Off-Line MF (Hz/s)	Error (Hz/s)
No. 1	−0.0165	−0.0165	0	−0.0078	−0.0075	−0.0003
No. 2	−0.0118	−0.0119	0.0001	−0.0160	−0.0157	−0.0003
No. 3	−0.0023	−0.0022	−0.0001	−0.0138	−0.0137	−0.0001
No. 4	−0.0197	−0.0198	0.0001	−0.0191	−0.0191	0
No. 5	−0.0015	−0.0013	−0.0002	−0.0032	−0.0035	0.0003
No. 6	−0.0157	−0.0156	−0.0001	−0.0189	−0.0184	−0.0005
No. 7	−0.0035	−0.0036	0.0001	−0.0130	−0.0132	0.0002
No. 8	−0.0241	−0.0241	0	−0.0063	−0.0058	−0.0005
No. 9	−0.0124	−0.0130	0.0006	−0.0158	−0.0162	0.0004
No. 10	−0.0305	−0.0304	−0.0001	−0.0158	−0.0162	0.0004
Sum	−0.0138 ± 0.00960	−0.0138 ± 0.00959	0.0000 ± 0.00024	−0.0130 ± 0.00544	−0.0129 ± 0.00546	−0.0000 ± 0.00037

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
