# Peer review of "An EMG Patch for the Real-Time Monitoring of Muscle-Fatigue Conditions During Exercise"

_sensors, 2019, doi:10.3390/s19143108_

Round 1
Reviewer 1 Report
The authors of the paper present a novel wearable EMG patch to monitor muscle fatigue. There are some flaws in the manuscript, which needs to be addressed:
1. The presentation of the manuscript is poor.
i. Rewrite the abstract and Conclusion.
ii. The method section confuses the reader. Please clearly explain the role of the accelerometer.
iii. Please write the conclusion and discussion separately.
2. The conclusion should justify the abstract.
3. The algorithm should be written in standard pseudo code format.
4. Please increase the font size of the figures.
5. Axes of Figure 5 are missing.
6. Please explain Figure 8 and Table 1 in clear words.
7. Result section needs more elaboration.
8. Please write some sentences about the privacy of the subjects.
9. The unit of power is not consistent in the manuscript.
10. What is the sampling frequency of the proposed system?
Reviewer 2 Report
Authors report a study on the design of an electromyography (EMG) patch that could be applied to monitor the muscle condition when doing an exercise. This EMG patch used a microcontroller unit (MCU) with the Cortex-M4 81 which calculate the median frequency (MF) of the first intrinsic component of empirical mode decomposition (EMD) for the sEMG in real time. They also developed an application program implemented in a smart phone to display the bicyclists’ muscle condition and the riding information in the real time. This is an interesting contribution to wearable devices. However, I have some suggestions :
· The wearable evaluation is not perfected and needs few confirmations. In my opinion, this is a preliminary study. The device needs to be integrated into belt… so that it could be considered as a wearable device.
· From Figure 6, tests performed here are with gel electrodes Ag/AgCl. Authors need to mention this in materials and methods. Also, the type of electrodes used to record EMG is extremely important because the design of their amplifier will depend on the contact impedance electrode-skin. In the case of gel electrodes authors did not encounter this problem. EMG signals will be degraded with time because the gel of Ag/AgCl electrodes dry out. These electrodes can also provoke some skin irritation.
Do authors change electrodes from subject to subject?
I suggest authors to test their device by using some textile electrodes such as silver plated electrodes commercially available.
· It could be more relevant if authors discuss some recent studies in introduction
· Line 26 : two electrodes circuit was designed to measure the EMG The power consumption à a point is missing
· Lines 113-115: Why the choices of the Butterworth digital filter and not another type? Why a second order exactly?
· Lines 121-122 : Is there a repetition ? ‘’The IC (TPS78233, TI) is used to provide a voltage of 3.3V’’
· Line 124 : ‘’(Nordic Semiconductor, UAS)’’. Do you mean USA ?
· Lines 225-228: this part need to be in Materials and Methods section
·
· Line 253 : Figure 10 :It could be better to put the Map in English.
· Line 259 : ‘’….rate et al. [20].’’ et al needs to be deleated.
· Line 263 : ‘’Therefore, for wearable device has higher’’ à Add that ‘’Therefore, for wearable device THAT has higher…’’
Reviewer 3 Report
In this paper, the authors present a wearable EMG device for real-time monitoring of muscle-fatigue condition during exercise. Although the idea might be interesting for this journal, the paper has many important and serious technical and non-technical lacks.
Major comments:
- Grammar and typos are usually minor comments but, in this case, there are a lot of them. These make the paper almost illegible and very poor in scientific soundness. Some examples are: use of present and past verb tenses in the same context, improper use of singular/plural verb tenses, improper use of determiners (e.g., a and the), use of imprecise words (e.g., issue), several typos (e.g., 500k Hz), etc. All this is easily noticeable by reading the abstract.
- The paper is badly written. Apart from English, sometimes I did not find a connecting thread between paragraphs.
- Abstract: Why is important to determine muscle fatigue? The need of the work must be in the abstract (as well as it is well-described in the introduction). What do the authors want to mean with "the first and second times"? The abstract should be re-written.
- Methods: A very important issue: the authors use band-pass filtering (30 - 500 Hz) at 1000 Hz of sampling rate. This is on the limit of Nyquist criterion. Taking into account that the filter has a slope, I am sure that there is aliasing in the recordings. A 0.25*Fs low-pass filtering is usually recommended to avoid aliasing. In addition, the experiemental protocol description is poor.
- Results: The authors try to validate (without any statistical test) a system for monitoring/detecting muscle fatigue by comparing the results of a signal acquisition + signal processing obtained using their system with the ones obtained using a commercial system + offline processing. This makes no sense. How do you know that you are monitoring fatigue without a gold standard (e.g., force sensor to measure fatigue)?
- Conclusions: They are not sufficiently supported by the results.
Minor comments:
- Some acronyms are not defined (e.g., ID, RPM or BLE)
- Please do not use ERMS for RMS errors. ERMS is usually used as the RMS value of the electric field
- Quality and resolution of pictures are generally poor.
Conclusion: This work has a good and interesting engineering work behind. However, it has grave scientific and technical flaws (apart from language issues) and it should not be published in this journal. I recommend that the authors design and conduct a new study and write a new paper in order to use all the good engineering work already done.
Round 2
Reviewer 1 Report
1. The authors are suggested to improve the empirical mode decomposition (EMD). 2. For better understanding of the readers, write the steps (1--9) of EMD in Pseudo code format.Author Response
Please see the attachment.

Reviewer 2 Report
Authors have improved the manuscript, however they don't really reply to my remarques in point 1 and 2. Tables added are not necessary. In fact, the title of the article ''A Wearable EMG Patch for the Real Time Monitoring 3 of Muscle-Fatigue Conditions during Exercise'' is not completely compatible with the study. Wearable EMG ... means that the (device +electrodes) need to be embedded into a textile so that everyone could wear it... For this reason, I ask again authors to think about a manner to integrate their device into a belt for instance by sewing.... Also, Ag/AgCl electrodes are not suitable for long-term monitoring. Even for ECG Holter, these electrodes are changed permanently that's why I ask again authors to test their device with textile electrodes (silver plated electrodes are commercially available).
Reviewer 3 Report
The authors have successfully addressed all the comments. However, I would change the sentence "Thus, the sampling EMG signal must have the aliasing phenomenon." in discussion. I would say "might/may have some aliasing".
Round 3
Reviewer 2 Report
Authors have improved their paper and I suggest it to be published. However, pictures in Figure 12 need to have the same size.